# Monitoring single-cell dynamics of entry into quiescence during an unperturbed life cycle

Basile Jacquel[1,2,3,4†], Théo Aspert[1,2,3,4†], Damien Laporte[5], Isabelle Sagot[5], Gilles Charvin[1,2,3,4]*

[1]Department of Developmental Biology and Stem Cells, Institut de Génétique et de Biologie Moléculaire et Cellulaire, Illkirch, France; [2]Centre National de la Recherche Scientifique, Illkirch, France; [3]Institut National de la Santé et de la Recherche Médicale, Illkirch, France; [4]Université de Strasbourg, Illkirch, France; [5]Institut de Biochimie et Génétique Cellulaires, UMR 5095 CNRS - Université de Bordeaux, Bordeaux, France, Bordeaux, France

**Abstract** The life cycle of microorganisms is associated with dynamic metabolic transitions and complex cellular responses. In yeast, how metabolic signals control the progressive choreography of structural reorganizations observed in quiescent cells during a natural life cycle remains unclear. We have developed an integrated microfluidic device to address this question, enabling continuous single-cell tracking in a batch culture experiencing unperturbed nutrient exhaustion to unravel the coordination between metabolic and structural transitions within cells. Our technique reveals an abrupt fate divergence in the population, whereby a fraction of cells is unable to transition to respiratory metabolism and undergoes a reversible entry into a quiescence-like state leading to premature cell death. Further observations reveal that nonmonotonous internal pH fluctuations in respiration-competent cells orchestrate the successive waves of protein superassemblies formation that accompany the entry into a *bona fide* quiescent state. This ultimately leads to an abrupt cytosolic glass transition that occurs stochastically long after proliferation cessation. This new experimental framework provides a unique way to track single-cell fate dynamics over a long timescale in a population of cells that continuously modify their ecological niche.

**\*For correspondence:**
charvin@igbmc.fr

†These authors contributed equally to this work

**Competing interest:** The authors declare that no competing interests exist.

## Editor's evaluation

The cell fate program that is set in motion as yeast cells transition from fermentation to respiration is still not well understood. The development of the microfluidic platform described in this manuscript could make a significant contribution to understanding the succession of metabolic and structural changes occurring during this transition. The application of single cell tracking to monitor the temporal program of these changes represents a major technical advance that will be of general interest to researchers interested in defining the developmental programs that contribute to cellular quiescence and longevity.

## Introduction

Microorganisms have evolved plastic growth control mechanisms that ensure adaptation to dynamical environmental changes, including those that arise from their proliferation (such as nutrients limitations and cellular secretion in the medium). During its natural life cycle, budding yeast may undergo several metabolic transitions from fermentation to respiration, followed by entry into a reversible state of

proliferation arrest known as quiescence (*De Virgilio, 2012*; *Gray et al., 2004*; *Sun and Gresham, 2021*; *Miles et al., 2021*). Despite quiescence being an essential part of the microorganism life cycle that ensures cell survival over prolonged periods (*Fontana et al., 2010*), it has received little attention compared to the analysis of biological processes in proliferative contexts.

Quiescent cells strongly differ from proliferating cells in terms of metabolic activity and gene expression (*Gray et al., 2004*; *Miles et al., 2013*). They also display a large body of structural rearrangements in the cytoskeleton, mitochondria, nuclear organization, and the appearance of protein superassemblies, clusters, and aggregates (*Sagot and Laporte, 2019*; *Sun and Gresham, 2021*). So far, the complex and entangled regulatory processes controlling this particular state's establishment remain poorly understood. In particular, the detailed sequence of events describing how the dynamics of metabolic cues during the natural life cycle drive the entry into quiescence is still missing.

Also, an essential feature of microbial ecosystems in the stationary phase (SP) is the existence of phenotypic variability (*Campbell et al., 2016*; *Avery, 2006*; *Holland et al., 2014*; *Labhsetwar et al., 2013*; *Ackermann, 2015*; *Bagamery et al., 2020*; *Solopova et al., 2014*) and history-dependent behaviors that lead to fate divergences (*Balaban et al., 2004*; *Cerulus et al., 2018*). In quiescence, the coexistence of heterogeneous cell populations has been previously reported (*Allen et al., 2006*; *Laporte et al., 2018a*). Nevertheless, how phenotypic diversity emerges in a clonal population during a natural life cycle remains elusive (*Figure 1—figure supplement 1A*). Bridging this gap requires performing longitudinal tracking of individual cells over time. However, an important technical obstacle is that it must be done in population-scale growth experiments to allow cell proliferation to have a collective impact on the environment.

Previous work has used an abrupt transition to glucose starvation to study how cells reorganize upon entry into the SP in various biological contexts (*Munder et al., 2016*; *Bagamery et al., 2020*). While this experimental framework may be helpful for studying standard properties of cells undergoing proliferation arrest, the results cannot be transposed to the context of entering quiescence during an undisturbed life cycle, in which cells undergo a sequence of metabolic transitions and feedback continuously into the composition of their environment. Indeed, an essential condition to reach a *bona fide* quiescence state (i.e., the ability to recover proliferation after prolonged arrest) is that cells must experience a respiration phase (RP) to accumulate carbohydrates, which does not occur upon abrupt glucose starvation (*Ocampo et al., 2012*; *Li et al., 2013*). In addition, different nutrient limitations lead to distinct quiescent states (*Klosinska et al., 2011*). Therefore, it is essential to develop novel methods that capture the true dynamics of cell transitions as they may occur in their ecological niche (*Miles et al., 2021*).

Here, we report the development of a microfluidic platform for single-cell ecology, allowing continuous tracking of individual cells' fate during an unperturbed full life cycle (up to 10 days). Using a fluorescent reporter of internal pH (*Mouton et al., 2020*; *Miesenböck et al., 1998*; *Dechant et al., 2010*; *Munder et al., 2016*), we observe that the diauxic shift (DS) witnesses a cell fate divergence, where a minority of cells experience a metabolic crash similar to that observed upon an abrupt starvation (*Bagamery et al., 2020*). Interestingly, our long-term tracking capabilities further reveal that these cells experience a premature yet reversible induction of cellular reorganizations that coincide with limited survival. In contrast, respiration-competent cells experience fluctuations in internal pH in sync with metabolic transitions that drive successive waves of cellular reorganizations and a stochastic switch to a glass transition of the cytoplasm long after proliferation cessation. Altogether, our analysis reveals how metabolic changes encountered by yeast cells during an unperturbed life cycle coordinate the temporal control of complex cellular reorganizations.

## Results

To track individual cell behavior during an unperturbed life cycle, we set up a device composed of a 25 ml liquid yeast culture (YPD medium) connected to a microfluidic device for single-cell observation (*Figure 1A*, *Figure 1—figure supplement 1B and C*, 1F). Thanks to a closed recirculation loop, individual cells trapped in the microfluidic device could be imaged over time while experiencing the same environmental changes as the population liquid culture. To prevent clogging in the microfluidic device due to the high cell density in the culture (up to $10^9$ cells/ml), we designed a filtration device based on inertial differential migration (*Kuntaegowdanahalli and Papautsky, 2008*). Using this technique, cells were rerouted back to the liquid culture before entering the microfluidic device (*Figure 1A*).

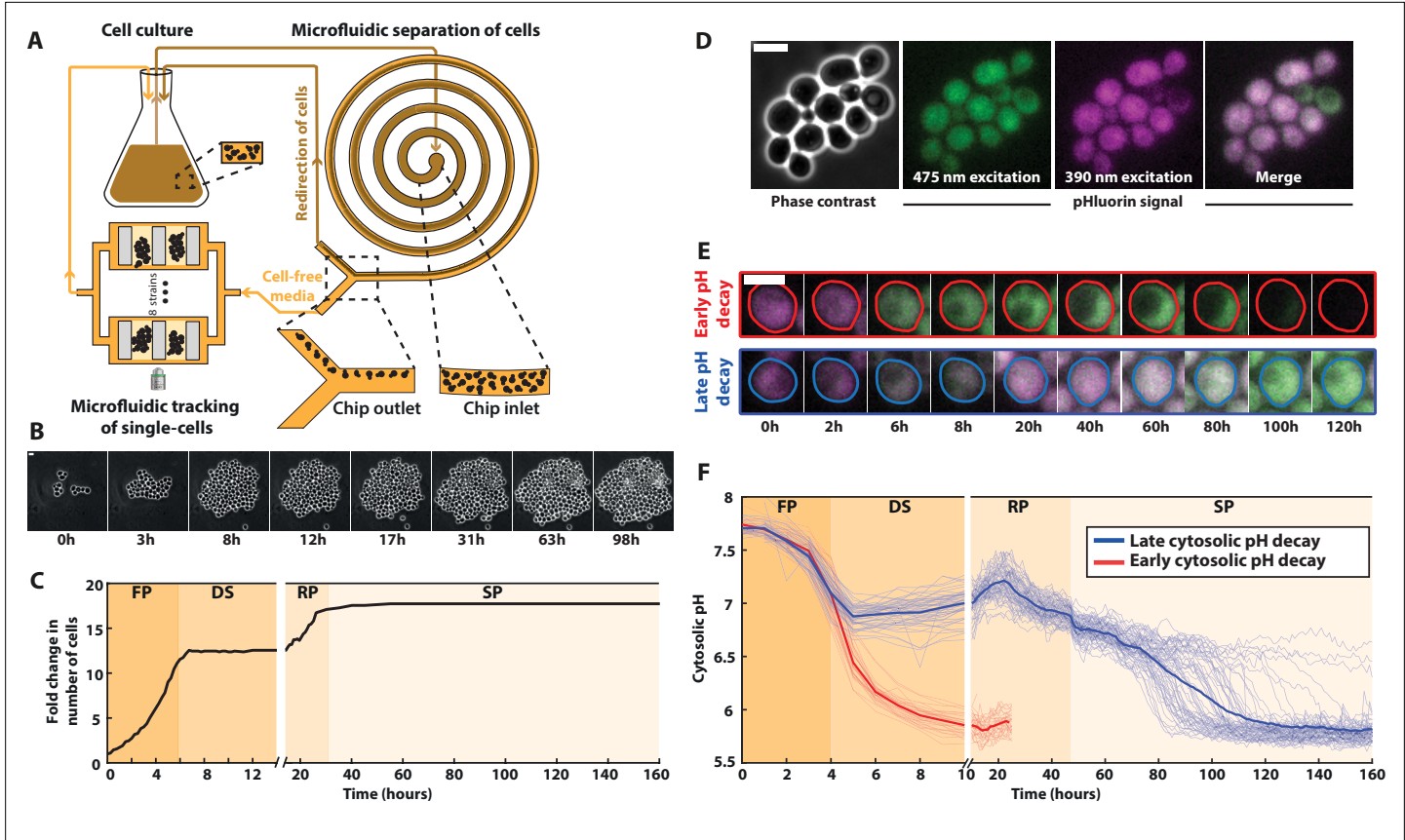

**Figure 1.** A microfluidic platform for single-cell tracking during the yeast proliferation cycle. (**A**) Schematics of the experimental setup, representing the liquid culture flask, the observation microfluidic device with trapped cells, and the microfluidic filtering device designed to redirect the cells back to the liquid culture while recirculating the medium of the liquid culture to the observation microfluidic chamber. (**B**) Sequence of phase-contrast images of cells growing in the microfluidic device. Scale bar = 5 μm. (**C**) Fold increase in cell number over an entire life cycle for the microcolony displayed in B; each shaded area represents a distinct proliferation phase, which was determined using piecewise linear fitting to cell proliferation data (see Materials and methods and *Figure 1—figure supplement 1G* for details): fermentation phase (FP), diauxic shift (DS), respiration phase (RP), and stationary phase (SP). (**D**) Cluster of cells showing typical phase-contrast, fluorescence, and overlay images using the cytosolic pH sensor pHluorin. (**E**) Typical sequences of overlaid fluorescence images obtained with the pHluorin sensor at indicated time points. Colored lines indicate cell contours. (**F**) Quantification of the absolute cytosolic pH as a function of time; each line represents an individual cell, while the bold line indicates the average among cells with either an early (red lines, *N* = 32 cells displayed) or late (blue line, *N* = 64 cells displayed) decaying pH.

The online version of this article includes the following video and figure supplement(s) for figure 1:

**Source data 1.** Spreadsheet containing the numerical values used to plot panels 1C and D.

**Figure supplement 1.** Design and calibration of the experimental setup.

**Figure supplement 2.** Population measurements of cell growth during an entire life cycle.

**Figure supplement 3.** Single-cell dynamics of entry into stationary phase in BY versus W303 strains.

**Figure 1—video 1.** Top: phase-contrast video of a microcolony growing in the observation device during the life cycle of the culture.
https://elifesciences.org/articles/73186/figures#fig1video1

**Figure 1—video 2.** Top: phase-contrast (left) and pHluorin (right, channels 1 and 2 merged as in Figure 1) video of a microcolony growing in the observation device during the life cycle of the culture.
https://elifesciences.org/articles/73186/figures#fig1video2

Optical density (OD) and fluorescence measurements revealed a filtration efficiency superior to 99%, reducing the concentration of cells entering the device by two orders of magnitude (*Figure 1—figure supplement 1*– S1E), and allowing us to image the cells over up to 10 days. We also checked that the same filtration device could sort *S. pombe* cells with 92% efficiency, thus highlighting the versatility of the methodology.

Using this methodology, we successfully recapitulated the successive proliferation phases occurring at the population level upon carbon source exhaustion with single-cell resolution (*Richards, 1928*), namely: a rapid exponential growth (doubling time = 84 ± 12 min) corresponding to glucose fermentation (referred to as the fermentation phase or FP in the following, from $t = 0$ to $t = 5.5$ hr), followed by a sharp growth arrest, or DS (from $t = 5.5$ to $t = 13.9$ hr); then, the resumption of a slow proliferative regime (doubling time = 307 ± 52 min) which is associated with the use of ethanol as a carbon source for a respiratory metabolism (RP, from $t = 13.9$ to $t = 31.6$ hr) and a final cell proliferation cessation occurring upon carbon source exhaustion, leading to SP, see *Figure 1B, C* and *Figure 1—video 1.Figure 1—source data 1* . To make sure that this growth pattern was not specific to the BY strain used for this experiment, we also made population growth measurements in various prototrophic and auxotrophic strains, and obtained similar results (*Figure 1—figure supplement 2*). To quantify the data further, the transition times between each metabolic phase were determined using piecewise exponential fits (*Figure 1—figure supplement 1* S1G). By refeeding the cells with fresh YPD medium after 10 days, we observed that up to ~80 % of them reentered the cell cycle within 5 hr (*Figure 1—figure supplement 1* -S1H). This result confirmed the reversibility of cell proliferation arrest and testified that cells establish *bona fide* quiescence in our growth conditions (*Laporte et al., 2018b*).

A drop in medium pH has long been reported to coincide with the resources' exhaustion during microbial growth (*Burtner et al., 2009*). Yet, how internal pH evolves over an entire life cycle has never been investigated. To address this, we used the ratiometric fluorescent probe of cytosolic pH, pHluorin (*Figure 1D*, *Miesenböck et al., 1998*; *Mouton et al., 2020*), which was calibrated to display the actual internal pH (*Figure 1D–F* and *Figure 1—figure supplement 1–* S1I). Using this readout, we observed that the pH, which was initially around 7.7, started to decline synchronously in all cells during the F phase (*Figure 1E, F* and *Figure 1—video 2*). At the onset of the DS, most cells (88%, *N* = 466) abruptly reached a plateau (pH ~6.9, blue lines on *Figure 1F*) followed by a slight pH increase (up to pH ~7.2) that coincided with entry into a respiratory metabolism. In contrast, a minority (12%, *N* = 466) of cells (*Figure 1E* and red lines on *Figure 1F*) experienced a further drop in pH down to about 5.8 during the DS. In this subpopulation, the fluorescence signal progressively disappeared, precluding monitoring the internal pH for more than 20 hr in a reliable manner.

In cells with high internal pH during the DS, the pH gradually declined after reaching a local maximum during the R phase. These cells then experienced a sharp pH drop down to about 5.8, which occurred at very heterogeneous times during the SP, unlike the cells with an early pH drop. Altogether, these observations revealed unprecedented dynamics of internal pH during the yeast life cycle: pH variations appeared to be in sync with the sequence of proliferation phases, suggesting that internal pH is a crucial marker of the cells' metabolic status during their life cycle.

These continuous pH measurements also unraveled a divergence in cell fate at the DS, leading to the early emergence of heterogeneity within the population, in line with previous observations made upon abrupt starvation (*Bagamery et al., 2020*). This phenomenon was also observed in a W303 strain (*Figure 1—figure supplement 3*). Recent studies have shown that the activation of respiration was a crucial metabolic response to survive glucose deprivation (*Weber et al., 2020*), enabling carbohydrate storage (*Ocampo et al., 2012*) and long-term viability (*Laporte et al., 2018a*). Interestingly, it was shown that a respiration defect was naturally observed in about 10 % of cells upon proliferation cessation following glucose exhaustion (*Laporte et al., 2018a*). Hence to further characterize whether differences in metabolic status drove the emergence of divergent cell fates at the DS, first, we quantified cellular proliferation over time using single-cell area measurements. We found that cells with a late pH drop resumed growth and roughly doubled their biomass during the R phase (blue lines on *Figure 2A* and *Figure 2—source data 1*) in agreement with *Figure 1C*.

In contrast, cells that experienced an early pH drop (red lines on *Figure 2A*) did not recover during the R phase, suggesting that they could not transition to respiratory metabolism. Indeed, these cells did not resume growth when adding lactate (i.e., a nonfermentable carbon source) to the medium neither (*Figure 2—figure supplement 1A*). However, most of them were still viable after 3 days in SP, even though their survival declined faster than adapting cells (*Figure 2B*).

Second, to assess cellular respiratory function, we used the mitochondrial Ilv3-mCherry marker (*Laporte et al., 2018a*). Whereas the mitochondrial network architecture appeared similar (i.e., tubular) in all cells before the DS, proliferating cells in the R phase displayed a fragmented mitochondria phenotype typical of respiring cells (blue cell, *Figure 2C*). In contrast, nonproliferating

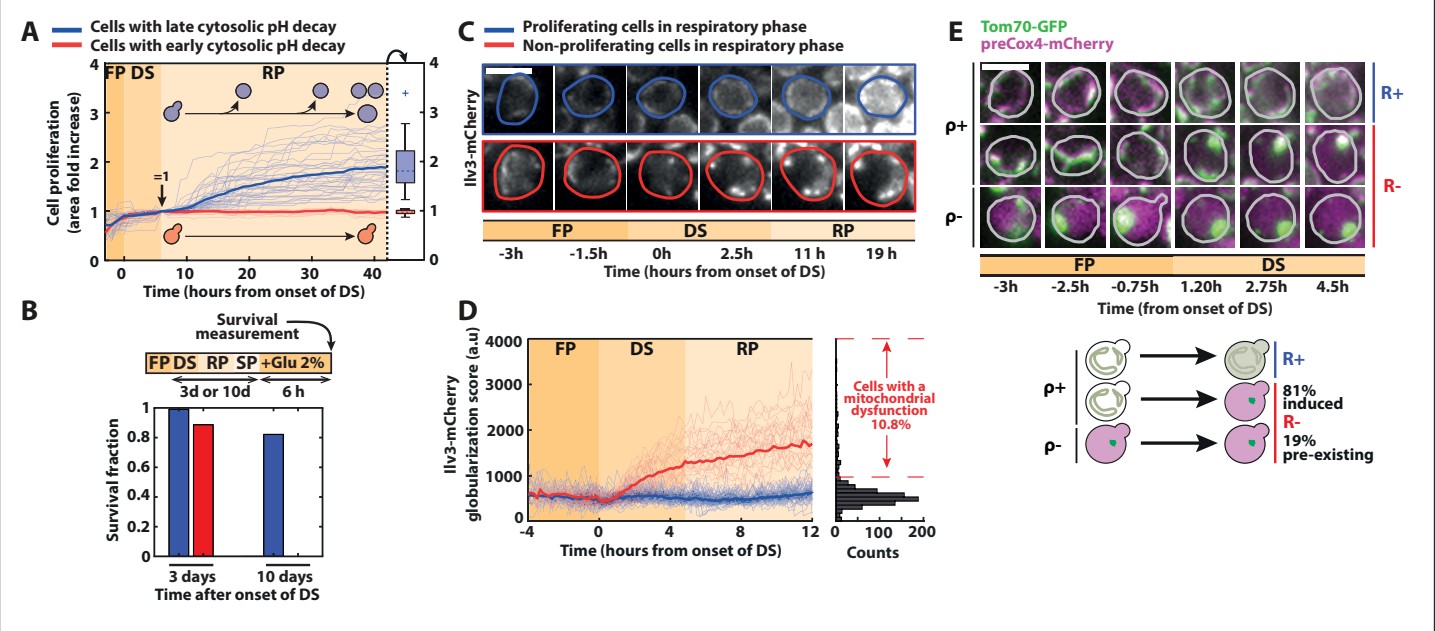

**Figure 2.** Divergent cell fates induced by a metabolic challenge at the diauxic shift (DS). (**A**) Quantification of single-cell growth during F, DS, and R phases, as defined in *Figure 1*. Each line represents the fold area increase (including buds) of single cells over time, normalized by cell area at the end of the DS (N = 50). The bold lines represent the averages over all the cells that experience fast (red) and slow (blue) pH decay, respectively. Right: box plot indicating the fold increase in cellular area in each subpopulation during the R phase (N = 40 for the slow pH decay population, N = 10 for the fast pH decay population). (**B**) Fraction of surviving cells among adapting (blue bars) and nonadapting (red bars) cells, measured by quantifying the cells' ability to resume growth 6 hr after reintroduction of fresh medium (2 % glucose) at 3 (N = 53 for red bars, N = 221 for blue bars) or 10 days (N = 114 for red bars, N = 403 for blue bars) after the DS. Scale bar = 5 µm. (**C**) Representative sequence of fluorescence images obtained with the Ilv3-mCherry mitochondrial marker at the indicated time points for the different classes of cells, as indicated by the colored contour. (**D**) Single-cell quantification of a globularization score (see Materials and methods) from fluorescence images over time for both adapting (blue line; N = 81) and nonadapting cells (red line; N = 28). The bold lines represent averages within each subpopulation. Right: histogram of globularization score for each cell (N = 720). (**E**) Preexisting versus newly occurring respiratory defects in cells experiencing the DS Sequence of fluorescent images (overlay of preCox4-mCherry and Tom70-GFP) at indicated times. Scale bar = 5 µm. Each line represents a different type of cell fates (top: $\rho$ + R+ ; middle: $\rho$ + R−; bottom: $\rho$ − R−). Schematics: representation of the different cell fates based on the fluorescence patterns of preCox4-mCherry and Tom70-GFP, with the quantification of the fraction of each subpopulation (N = 701).

The online version of this article includes the following video and figure supplement(s) for figure 2:

**Source data 1.** Spreadsheet containing the numerical values used to plot panels 2A and D.

**Figure supplement 1** Complementary analyses of the divergent cell fate at the diauxic shift.

**Figure 2—video 1.** Phase-contrast (left) and Ilv3-mCherry (right) video of a microcolony growing in the observation device during the life cycle of the culture.
https://elifesciences.org/articles/73186/figures#fig2video1

cells underwent a globularization of their mitochondrial network (red cell, *Figure 2C* and *Figure 2— video 1*). We further quantified the mitochondrial network's reorganization dynamics by computing a custom aggregation index that discriminates globularized versus tubular and fragmented mitochondria (*Figure 2D* and *Figure 2—source data 1*, see Materials and methods for details). Based on the clear distinction in the aggregation index between adapting and nonadapting cells, we measured that about ~10 % of the cells could not transition to a respiratory metabolism (at *t* = 12 hr post-DS, *Figure 2D*), in agreement with previous findings (*Laporte et al., 2018a*). Importantly, this quantification revealed that the mitochondrial globularization in nonadapting cells was temporally closely associated with the DS since it started as early as 1h20 (p < 0.05) after its onset (*Figure 2C,D*). Altogether, these results demonstrate that proliferating and nonproliferating cells experienced divergent cell fates at the DS based on their ability to switch to respiration; hence, they were referred to as respiration positive (R+) and negative (R−), respectively. Also, these results suggest that the inability of R− cells to activate a respiratory metabolism was either triggered by this metabolic challenge or, alternatively, preexisted the DS, knowing that respiratory deficient cells (i.e., $\rho$ − cells) are common in

the BY background (yet less so in W303 strains) due to the genetic instability of mitochondrial DNA (*Dimitrov et al., 2009*).

To discriminate between these two hypotheses, we used the mitochondrial localization marker Tom70-GFP (Tom70 is a protein of the outer mitochondrial membrane) and a preCox4-mCherry fusion (preCox4 is a nuclear-encoded mitochondrial protein that is imported only in functional mitochondria *Veatch et al., 2009*), to assess the cells' ability to respire (*Fehrmann et al., 2013*). Using these markers, we first checked that all R+ cells maintained a functional preCox4-mCherry import from fermentation to respiration (i.e., they were $\rho$ + cells, *Figure 2E*). Then, we observed that among the R− cells, only a minority (i.e., 19%) had a dysfunctional mitochondrial import before the DS, indicating that they had a preexisting respiratory deficiency (see $\rho$ − cells in *Figure 2E*). In contrast, the vast majority of R− cells (81%) transitioned from $\rho$ + to $\rho$ − during the DS. This result demonstrates that the respiration defect observed in R− cells occurred concomitantly with the environmental switch and was therefore not due to a preexisting condition. Also, we showed that the progeny quite faithfully inherited it (inheritance index = 0.52, i.e., is smaller than 1, see *Figure 2—figure supplement 1B* and Material and methods for details). Altogether, these observations supported a scenario in which a fate divergence occurred early at the onset of the DS, where R+ cells quickly switched to a respiratory metabolism, while R− cells failed to do so.

As numerous cellular reorganizations occur during entry into quiescence (*Sagot and Laporte, 2019*), we sought to quantitatively determine how they are coordinated with metabolic transitions during the life cycle. Indeed, the metabolism controls internal cellular pH, which in turn can induce significant physicochemical changes in the proteome, such as protein aggregation and phase transition (*Munder et al., 2016*; *Dechant et al., 2010*). To do this, we monitored the dynamics of formation of supramolecular bodies associated with quiescence or the response to starvation: P-bodies formation (using the Dhh1-GFP fusion *Beckham et al., 2008*; *Balagopal and Parker, 2009*; *Mugler et al., 2016*), metabolic or regulatory enzymes prone to aggregation (Gln1-GFP and Cdc28-GFP) (*Narayanaswamy et al., 2009*; *Petrovska et al., 2014*; *Shah et al., 2014*), actin bodies formation (Abp1-GFP) (*Sagot et al., 2006*), and proteasome storage granules (PSGs, using the Scl1-GFP fusion) (*Laporte et al., 2008*; *Peters et al., 2013*).

We found that the cellular reorganizations were highly coordinated with the sequence of metabolic phases and the cellular proliferation status (*Figure 3A* and *Figure 3—source data 1*): in R+ cells, Dhh1-GFP and Gln1-GFP (*Figure 3—video 1*) foci appeared at the onset of the DS, then were partially dissolved during the R phase, and reappeared upon entry into SP. Other hallmarks of proliferation cessation, such as actin bodies, PSGs, and other protein foci (Cdc28-GFP), showed up only at the end of the R phase. Importantly, we observed that R− cells also experienced a consistent formation of fluorescent foci for the markers that we monitored, yet, unlike R+ cells, they all appeared during the DS. In addition, all these foci ultimately disappeared, presumably as a consequence of the premature cell death observed in this subpopulation.

By overlapping the dynamics of fluorescence foci formation with that of internal pH (from *Figure 1F*), we noticed that the formation of bodies overall coincided with variation in absolute pH level in both R+ and R− cells, even though it appeared at different timescales. This finding is compatible with the hypothesis that the drop in internal pH – which reflects a decrease in metabolic activity – actually drives successive waves of body formation and the appearance of quiescence hallmarks. Indeed, upon the DS, a pH decrease down to ~7 would trigger the formation of Gln1 and Dhh1 foci in both R+ and R− cells. In R+ cells, the rerise of the pH associated with proliferation resumption in the R phase would induce the disassembly of these structures until the pH reaches 7 again, after carbon source exhaustion in SP. In both R+ and R−, actin bodies, PSG, and Cdc28 foci would form only when the intracellular pH reaches 6.

To further check this hypothesis, since energy depletion was shown to induce a global modification of the cytoplasm to a glassy state (*Munder et al., 2016*), we sought to observe these transitions by monitoring the frame-to-frame displacement of Gln1-GFP foci over time (*Figure 3B*, *Munder et al., 2016*). This marker was chosen because foci are already present at the DS. We found that the mobility of Gln1-GFP foci decreased sharply in R− cells during the DS (*Figure 3—video 1*) and much later in R+ cells, consistently with the differences observed regarding the times of pH drops in both populations (*Figure 3C*). To better establish the links between pH and mobility, we exploited the fact that lipid droplets (LDs) could be conveniently observed using phase-contrast images (*Heimlicher*

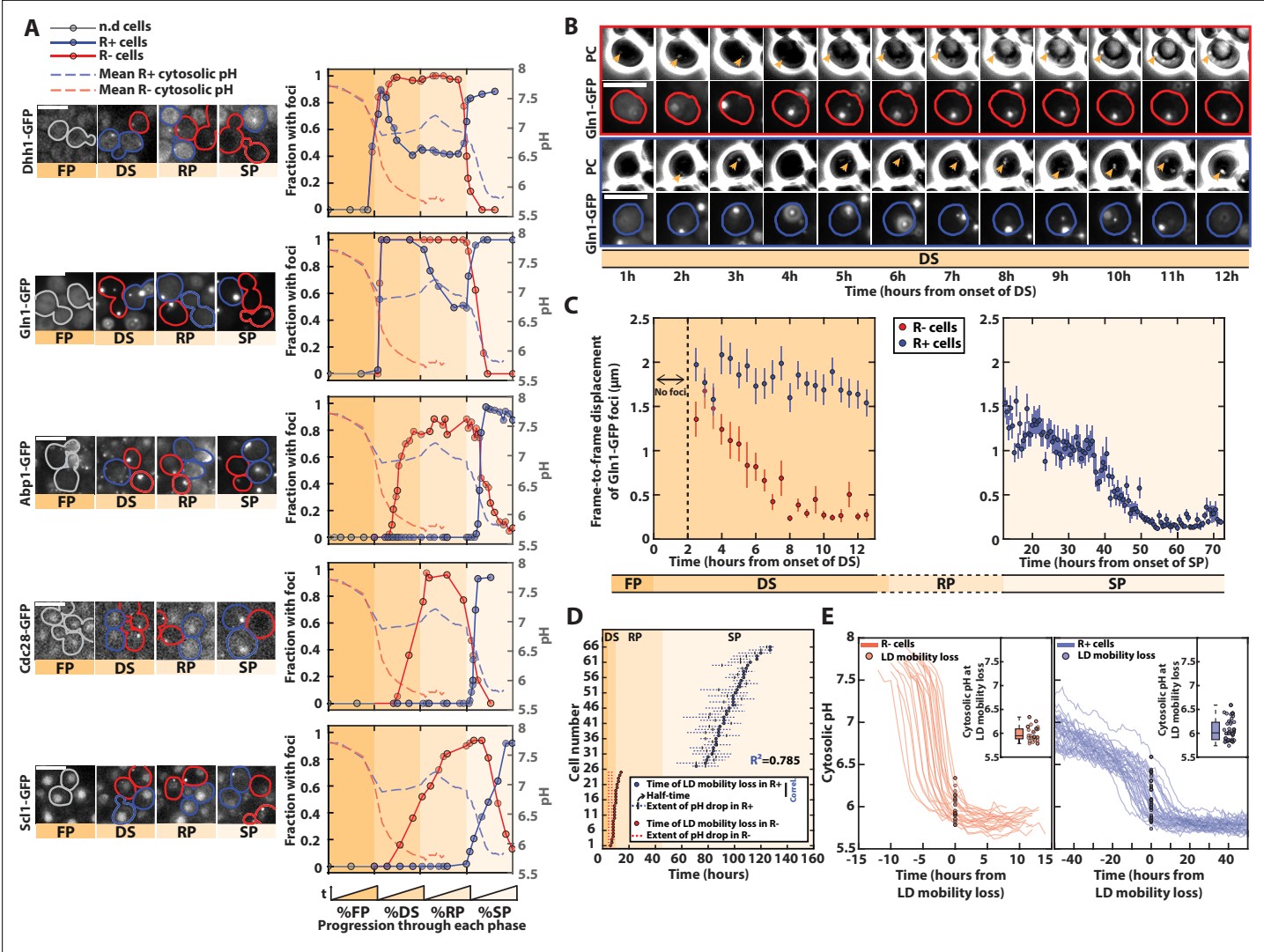

**Figure 3.** pH-driven phase transition to a gel-like state upon proliferation cessation. (**A**) Observation and quantification of fluorescent foci formation for the indicated fusion proteins (Dhh1-GFP, Gln1-GFP, Abp1-GFP, Cdc28-GFP, and Scl1-GFP). Left: each strip of fluorescence images displays unrelated cells at different phases during entry into stationary phase (SP). Colored contours indicate cells of interest (red for R− cells, blue for R+ cells, gray before the diauxic shift [DS]). Right: quantification of the fraction of cells with foci for each indicated fluorescent marker, as a function of the normalized time spent in each phase. Each solid colored line represents an indicated subpopulation of cells. The dashed colored lines represent the evolution of pH over time, based on data obtained in *Figure 1*; $N > 25$ cells for each marker. Scale bar = 5 µm. (**B**) Mobility of Gln1-GFP fluorescent foci and observation of lipid droplets (LDs). Sequence of phase-contrast and Gln1-GFP fluorescence images at indicated time points. The colored contours indicate the cells of interest (red and blue for R− and R+ cells, respectively). The orange arrowheads on the phase-contrast images indicate the LD. Scale bar = 5 µm. (**C**) Mobility of Gln1-GFP fluorescent foci. Quantification of average frame-by-frame displacement of Gln1-GFP foci for R+ and R− cells (blue and red points, respectively) starting after the appearance of foci ($t > 2$ hr after the onset of DS). Error bars represent the standard error on mean ($N = 51$ for R− and $N = 110$ for R+). The right plot only features R+ cells, since Gln1-GFP foci are no longer present in the SP phase in R− cells ($N = 28$). (**D**) Temporal link between the drop in internal pH and the time of mobility loss of LD ($N = 25$ for R− and $N = 43$ for R+). Each line corresponds to a single cell and represents the extent of pH drop (see Material and methods for details). Half-times of these drops are represented by a small vertical bar. The time of LD mobility loss is displayed as a dot. Of note, two cells did not display a pH drop, nor an LD mobility loss, hence they are not displayed on the plot. (**E**) Measurement of internal pH at the time of LD mobility loss. Overlay of internal pH in single cells obtained after synchronizing all traces with respect to the time of LD mobility loss, for R− (left) and R+ cells (right). Each dot represents the pH at the time of mobility loss in each cell ($N = 25$ for R− and $N = 43$ for R+). Inset: box plot showing the distribution of pH values of every single cell at the time of LD mobility loss.

The online version of this article includes the following video and figure supplement(s) for figure 3:

**Source data 1.** Spreadsheet containing the numerical values used to plot panel 3A.

**Source data 2.** Spreadsheet containing the numerical values used to plot panels 3D and E for R+ cells.

*Figure 3 continued on next page*

*Figure 3 continued*

**Source data 3.** Spreadsheet containing the numerical values used to plot panels 3D and E for R− cells.

**Figure supplement 1** Coincidence of lipid droplets and Gln1-GFP foci mobility.

**Figure 3—video 1.** Phase-contrast (left) and Gln1-GFP (right) video of a microcolony growing in the observation device from fermentation phase to respiration phase.

https://elifesciences.org/articles/73186/figures#fig3video1

*et al., 2019*). Similarly, the time of loss of LDs mobility correlated very well with that of Gln1-GFP foci (*Figure 3—figure supplement 1A*). By quantifying both the time of loss of LD mobility and the dynamics of pH drop in single cells, we observed that both events were tightly correlated in all cells, despite the large cell-to-cell heterogeneity in the time of pH drop in R+ cells (*Figure 3D* , *Figure 3—figure supplement 1B*, *Figure 3—source data 2* and *Figure 3—source data 3*). Also, after synchronizing all single-cell trajectories from the time of loss of LD mobility, we showed that the mobility loss of the cytoplasm occurred at a similar pH ~6 in both R+ and R− cells (*Figure 3E*, Figure 3—source data 2 and *Figure 3—source data 3*). Therefore, these observations suggest that the internal pH, which displays a very dynamic behavior during the yeast life cycle, induces waves of cellular structural remodeling that ultimately triggers a global transition of the cytoplasm to a gel-like state. This model is further supported by the concomitance between the premature pH drop and the phase transition in respiratory deficient (R−) cells. Altogether, it suggests that, upon nutrient exhaustion, cells undergo stereotypical pH-dependent structural reorganizations, no matter their respiratory status. R− cells' inability to switch to respiration, which ultimately compromises the long-term viability in this subpopulation (*Ocampo et al., 2012*; *Weber et al., 2020*), makes this transition precocious, and explains the divergent cell fates in the population at the DS.

## Discussion

In this study, we have developed a new microfluidic platform that allows us to monitor a full cell proliferation cycle in liquid culture with single-cell resolution. Individual cell tracking and quantitative fluorescence measurements provide a unique dynamic assessment of the successive metabolic transitions from fermentation to the SP observed in a liquid culture submitted to nutrients exhaustion. In contrast to most previous studies that used abrupt environmental switches to investigate the metabolic response to starvation, our methodology recapitulates the unperturbed dynamics of nutrients experienced by cells during a life cycle in laboratory conditions. We envision that this methodology could be further applied to other contexts in which collective cell behavior impacts the environment, which in turn shapes individual cellular responses – for example, metabolic oscillations (*Tu et al., 2005*) and cooperative behaviors (*Dal Co et al., 2020*; *Campbell et al., 2015*).

Continuous monitoring of cell growth and mitochondrial markers revealed a clear divergence in cell fate, where a minority of them failed to establish a respiratory metabolism, in line with recent observations obtained using an abrupt medium change (*Bagamery et al., 2020*). The dramatic drop in internal pH upon proliferation cessation in R− cells at the DS suggests that the failure to transition to respiration induces a major loss of energy homeostasis in this subpopulation, that may, in turn, compromise long-term viability (*Ocampo et al., 2012*; *Weber et al., 2020*). By and large, this phenomenon could not be explained by preexisting phenotypic differences in the population during the FP (*Bagamery et al., 2020*) but rather appeared to be triggered by the metabolic challenge associated with the exhaustion of glucose.

Previous studies have unraveled how changes in cytosolic pH – including upon nutrient exhaustion – control protein supramolecular assemblies (*Peters et al., 2013*; *Petrovska et al., 2014*). Our analysis shows that nonmonotonous fluctuations in pH level occur in sync with metabolic transitions during an unperturbed life cycle. Further, it reveals that successive pH drops are closely temporally related to the formation of many protein bodies and granules, despite the large cell-to-cell temporal variability associated with the onset of these events. Interestingly, since the structural reorganization of many protein complexes is one of the well-described hallmarks of quiescence (*Sagot and Laporte, 2019*), our observations thus suggest that internal pH acts as a key controller that drives waves of structural changes at well-defined pH levels during entry into quiescence. That R− cells display foci formation

similarly as the R+ cells provide further support to a model of stereotypical structural reorganizations driven by energy depletion and transduced by internal pH level.

In addition to monitoring the superassembly of specific markers, we showed that the transition to SP is accompanied by a transition of the cytoplasm to a glassy-like phase, hence transposing previous observations (*Parry et al., 2014*; *Joyner et al., 2016*; *Munder et al., 2016*) to the context of an unperturbed life cycle in budding yeast (*Heimlicher et al., 2019*). Importantly, the appearance of the glassy-like state occurs right at the DS in the R– cells and is significantly delayed and variable in time in R+ cells. We propose that the start of respiration upon glucose exhaustion prevents a precocious glass transition, which might be detrimental to cell viability if it occurs in an uncoordinated manner with other cellular reorganizations processes (e.g., energy storage)(*Ocampo et al., 2012*; *Weber et al., 2020*).

The abrupt and concomitant transition of all cells during the DS shows how the rapid evolution of the environment at this precise moment drives cell behavior in a deterministic manner, in the same way as during abrupt starvation (*Bagamery et al., 2020*; *Munder et al., 2016*). Conversely, the remarkable cell-to-cell temporal heterogeneity during the transition to a glassy state in R+ cells suggests that the cells have a developmental program whose progression is partly stochastic, that is, not entirely determined by the external environment. This last observation further supports that it is impossible to follow the process of entry into quiescence faithfully by imposing the dynamics of environmental changes (*Miles et al., 2021*) or based on population measurements only. Further studies using our methodology may allow us to discover how the succession of the different key steps of this developmental process contributes to establishing the specific physiological properties of quiescent cells (e.g., long-term survival and stress tolerance).

# Materials and methods

**Key resources table**

| Reagent type (species) or resource | Designation | Source or reference | Identifiers | Additional information |
|---|---|---|---|---|
| Strain, strain background (*S. cerevisiae*, BY, mat a) | WT | Euroscarf; PMID:9483801 | | |
| Strain, strain background (*S. cerevisiae*, BY, mat a) | BJQ-3 | Thermo Fisher; PMID:14562095 | | |
| Strain, strain background (*S. cerevisiae*, BY, mat a) | BJQ-7 | Thermo Fisher; PMID:14562095 | | |
| Strain, strain background (*S. cerevisiae*, BY, mat a) | BJQ-28 | Thermo Fisher; PMID:14562095 | | |
| Strain, strain background (*S. cerevisiae*, BY, mat a) | BJQ-23 | Thermo Fisher; PMID:14562095 | | |
| Strain, strain background (*S. cerevisiae*, BY, mat a) | BJQ3-3 | Thermo Fisher; PMID:14562095 | | |
| Strain, strain background (*S. cerevisiae*, BY, mat a) | Y10794 | Sagot Lab; PMID:18504300 | | |
| Strain, strain background (*S. cerevisiae*, BY, mat a) | YSF120-9D | Charvin Lab; PMID:24332850 | | |
| Strain, strain background (*S. cerevisiae*, BY, mat a) | SMY12 | Veenhoff lab; PMID:32990592 | | |
| Strain, strain background (*S. cerevisiae*, BY, mat a) | BJ2-44 | Thermo Fisher; PMID:14562095 | | |
| Strain, strain background (*S. cerevisiae*, BY, mat alpha) | Y11453 | This paper | | Results from a cross between Y11314 (Daignan-Fornier lab) and Y11453 |
| Strain, strain background (*S. cerevisiae*, S288C, mat alpha) | Y2658 | Daignan-Fornier Lab; PMID:19795422 | | |
| Strain, strain background (*S. cerevisiae*, FY, mat a) | Y2438 | Daignan-Fornier Lab; PMID:7762301 | | |

*Continued on next page*

*Continued*

| Reagent type (species) or resource | Designation | Source or reference | Identifiers | Additional information |
|---|---|---|---|---|
| Strain, strain background (*S. cerevisiae*, FY, mat a) | Y2439 | Daignan-Fornier Lab; PMID:7762301 | | |
| Strain, strain background (*S. cerevisiae*, FY, mat a/alpha) | Y12322 | This paper | | Results from a cross between Y2438 and Y2439 |
| Strain, strain background (*S. cerevisiae*, FY, mat a) | Y5738 | Sagot Lab; PMID:30299253 | | |
| Strain, strain background (*S. cerevisiae*, W303, mat a) | Y8037 | Sagot Lab; PMID:24338369 | | |
| Strain, strain background (*S. cerevisiae*, BY, mat a) | Y6735 | Sagot Lab; PMID:24338369 | | |
| Strain, strain background (*S. cerevisiae*, W303, mat alpha) | BJQ-11 | Alberti Lab; PMID: PMID:27003292 | | |

## Strains

All strains used in this study are congenic to BY4741 (see *Supplementary file 1* for details), unless specified otherwise (*Figure 1—figure supplement 2* and *Figure 1—figure supplement 3*).

## Cell culture

Freshly thawed cells were grown overnight. In the morning, 2 ml of the culture was inoculated into a 25 ml flask containing fresh YPD medium. After 5 hr, 2 ml of culture was used to load the cells into the microfluidic device, and the rest was used as circulating media for the experiment. This 5 hr delay was chosen so that cells only spend two to three divisions in a FP to limit the number of cells in the microfluidic device at the DS.

## Microfluidics and microfabrication

### Microfabrication

Microfluidic chips were generated using custom-made microfluidic master molds. The master molds were made using standard soft-photolithography processes using SU-8 2025 photoresist (Microchem, USA). The designs (which are available for download on github: https://github.com/TAspert/Continuous_filtration (*Aspert, 2021*) copy archived at swh:1:rev:8476f782bf6ff8fff2a9c78172cc8f072cc73916) were made on AutoCAD (Autoddesk, USA) to produce chrome photomasks (jd-photodata, UK). The observation device was taken from a previous study (*Goulev et al., 2017*).

The mold of the dust filter chip (see below for details) was made by spin coating a 25 μm layer of SU-8 2025 photoresist on a 3″ wafer (Neyco, FRANCE) at 2700 rpm for 30 s. Then, we used a soft bake of 7 min at 95 °C on heating plates (VWR) followed by exposure to 365 nm UVs at 160 mJ/cm² with a mask aligner (UV-KUB3 Kloé, FRANCE). Finally, a postexposure bake identical to the soft bake was performed before development using SU-8 developer (Microchem, USA).

The mold for the spiral-shaped cell filter device (see below for details) was obtained by spinning SU-8 2025 at 1750 rpm to achieve a 50 μm deposit. Bakes were 6 min long at 95 °C and UV exposure was done at 180 mJ/cm². A hard bake at 150 °C for 15 min was then performed to anneal potential cracks and to stabilize the resist.

Finally, the master molds were treated with chlorotrimethylsilane to passivate the surface.

### Microfluidic chip fabrication

The microfluidic devices were fabricated by pouring polydimethylsiloxane (PDMS, Sylgard 184, Dow Chemical, USA) with its curing agent (10:1 mixing ratio) on the different molds. The chips were punched with a 1 mm biopsy tool (Kai medical, Japan) and covalently bound to a 24 × 50 mm coverslip using plasma surface activation (Diener, Germany). The assembled chips were then baked for 1 hr at 60 °C to consolidate covalent bonds between glass and PDMS. Then, the dust filter chip was connected to the spiral cell filter which was in turn connected to the observation chip. All the connections used 1 mm (outside diameter) Polytetrafluoroethylene (PTFE)tubing (Adtech Polymer Engineering, UK).

All medium flows were driven using a peristaltic pump (Ismatec, Switzerland) at a 100 µl/min rate. The system was connected to the tank of media and cells described in the previous section. The observation chip was loaded with the cells of interest using a 5 ml syringe and a 23 G needle. Last, we plugged the cell outlet of the spiral and the observation chip outlet into the tank so the system is closed and without loss of media or cells.

## Microfluidic cell filter and dust filter

The spiral-shaped microfluidic device (*Kuntaegowdanahalli and Papautsky, 2008*) was designed to filter out cells coming from the liquid culture to prevent clogging in the observation device (*Figure 1— figure supplement 1C*). The dimensions of the cell filter (i.e., a channel of 100 µm width and 50 µm height, defining a spiral of five loops separated by 900 µm) were set to maximize the separation of haploid yeast cells, that is, ~5 µm particles, according to the following principle: particles in a spiral microfluidic channel with a rectangular section are submitted to several inertial forces that depend on their size and are either directed towards the center of the channel or the walls. Therefore, particles of similar diameters reach an equilibrium position and tend to focus on a single line, allowing their separation from the rest of the fluid by splitting the output channel in two different outlets. A similar filter could be used with other microorganisms by adapting the dimensions.

We also added a particle filter before the spiral to avoid any clogging of the spiral because of dust particles or debris (*Figure 1—figure supplement 1B*).

To measure the filtration efficiency of the cell filter, the cell concentration of the inlet and the two outlets was measured at four different time points (0 , 24 , 48 , and 120 hr) using a turbidity measurement (OD 660 nm, Fisherbrand). Another independent measurement was done using Green Fluorescent Protein (GFP) fluorescent yeast cells and measuring the fluorescence along the section of the device (See *Figure 1—figure supplement 1*). The filtration efficiency was equal to 99 % in both measurements, independently of the inlet cell concentration.

## Microscopy

For all experiments except pH measurements, cells in the observation device were imaged using an inverted widefield microscope (Zeiss Axio Observer Z1). Fluorescence illumination was achieved using LED lights (precisExcite, CoolLed) and the light was collected using a ×63 (N.A. 1.4) objective and an EM-CCD Luca-R camera (Andor). Standard GFP and mCherry filters were used.

For experiments using the pHluorin cytosolic pH probe, a Nikon Ti-E microscope was used along with a LED light (Lumencor) fluorescence illumination system. The fluorescence was measured using two excitation wavelengths using a standard roGFP2 filter set (AHF, peak excitation wavelengths 390 /18 and 475/28 nm, beamsplitter 495 nm, and emission filter 525/50 nm). Emitted light was collected using a ×60 N.A. 1.4 objective and a CMOS camera (Hamamatsu Orca Flash 4.0).

We used motorized stages to follow up to 64 positions in parallel throughout the experiment. Single plane images were acquired every 15 , 30 , 60 , or 240 min depending on the phase of the culture (high sampling rate during FP versus lower acquisition frequency in SP) to limit photodamage.

## Image processing and data quantification

### Calibration of the cytosolic pH probe

To calibrate the probe, 2 ml of exponentially growing culture (0.5 OD600) were centrifuged and resuspended in 200 µl calibration buffer (50 mM 2-(*N*-morpholino)ethanesulfonic acid (MES), 50 mM 4-(2-hydroxyethyl)-1-piperazineethanesulfonic acid (HEPES), 50 mM KCl, 50 mM NaCl, and 200 mM NH4CH3CO2) at various pH from 5 to 8, and supplemented with 75 µM monensin, 10 µM nigericin, 10 mM 2-deoxyglucose, and 10 mM NaN3, as described previously (*Mouton et al., 2020*). The cells were incubated in this buffer for 30 min and then imaged in the microfluidic device to perform a ratiometric fluorescence measurement (*Figure 1—figure supplement 1I*).

### Image processing

The raw images were processed using Matlab-based software (PhyloCell) and custom additional routines (*Fehrmann et al., 2013*; *Paoletti et al., 2016*). This software has a complete graphical interface for cell segmentation (based on a watershed algorithm), tracking (assignment cost minimization), and fluorescence signal quantification. The software can be downloaded online (https://github.com/

gcharvin/phyloCell*Charvin, 2021*). Raw data related to *Figure 1* can be found online: https://doi.org/105281/zenodo5592983.

### Data replicates
All measurements reported in this study are based on at least two replicates.

### Quantification of growth rate
The growth rate of the cells was computed using the evolution of the area of segmented cells (and buds) over time.

### Identification of growth phases
To determine the limit between successive growth phases, we used a piecewise linear fit to the evolution of the total number of cells with time (*D'Errico, 2017*). The time point of the intersection between two pieces of the piecewise linear fit was defined as the time of transition between the two corresponding phases (*Figure 1—figure supplement 1G*).

### Globularization score
The Ilv3-mCherry globularization score was measured in each cell by calculating the mean intensity of the five brightest pixels of the cell minus that of the other pixels, as previously described (*Cai et al., 2008*).

### Inheritance of the respiration defect
To quantify the inheritance of the respiration defect, we measured the average standard deviation of the respiration status (after assigning a value of 0 if the cells are unable to respire and one if they are respiration competent) within individual microcolonies of cells (microcolony size from 10 to 20 cells). We then obtained an 'inheritance index' by normalizing this value to the standard deviation obtained after taking all cells into account (i.e., with no distinction about their belonging to a given microcolony). Hence this index equals 1 when the respiration defect appears at random in lineages and 0 if the transmission of the phenotype is fully inheritable.

### Quantification of fluorescence foci
The fraction of cells displaying foci of GFP was manually scored at different time points using the ImageJ Cell Counter plugin.

### Mobility of fluorescent foci
The fluorescent foci in each cell were detected using the centroid position of the five brightest pixels. The frame-to-frame displacement of the foci was computed by iterating this procedure over all the frames and then averaged over a population of cells.

### Time of foci mobility loss and LDs
The time of mobility loss was determined by visual inspection of successive images due to the low signal-to-noise ratio in the image.

### Extend of pH drop
The start and end of the pH drops were determined using a piecewise linear adjustment, similar to what was used to determine the transition between metabolic phases.

## Acknowledgements

We thank Sara Mouton and Liesbeth Veenhoff for discussions and for sharing the pHluorin strain ahead of their publication, Audrey Matifas for constant technical support throughout this work, Titus Franzmann for fruitful discussions, Sophie Quintin and Sandrine Morlot for careful reading of the manuscript. We thank Denis Fumagalli at the Mediaprep facility. This work was supported by the Fondation pour la Recherche Médicale (FRM, BJ, and GC), the Agence Nationale pour la Recherche (TA and

GC), the grant ANR-10-LABX-0030-INRT, a French State fund managed by the Agence Nationale de la Recherche under the frame program Investissements d'Avenir ANR-10-IDEX-0002-02.

## Additional information

### Funding

| Funder | Grant reference number | Author |
|---|---|---|
| Fondation pour la Recherche Médicale | | Basile Jacquel Gilles Charvin |
| Agence Nationale de la Recherche | | Théo Aspert Gilles Charvin |

The funders had no role in study design, data collection and interpretation, or the decision to submit the work for publication.

### Author contributions

Basile Jacquel, Conceptualization, Data curation, Formal analysis, Investigation, Methodology, Software, Validation, Writing – original draft, Writing – review and editing; Théo Aspert, Conceptualization, Data curation, Formal analysis, Investigation, Methodology, Resources, Software, Validation, Writing – original draft, Writing – review and editing; Damien Laporte, Conceptualization, Investigation, Methodology, Resources, Writing – review and editing; Isabelle Sagot, Conceptualization, Methodology, Supervision, Writing – review and editing; Gilles Charvin, Conceptualization, Formal analysis, Project administration, Software, Writing – original draft, Writing – review and editing

### Author ORCIDs

Basile Jacquel (iD) http://orcid.org/0000-0002-5670-4635
Théo Aspert (iD) http://orcid.org/0000-0003-2957-0683
Damien Laporte (iD) http://orcid.org/0000-0002-1556-5253
Isabelle Sagot (iD) http://orcid.org/0000-0003-2158-1783
Gilles Charvin (iD) http://orcid.org/0000-0002-6852-6952

### Decision letter and Author response

Decision letter https://doi.org/10.7554/eLife.73186.sa1
Author response https://doi.org/10.7554/eLife.73186.sa2

## Additional files

### Supplementary files
• Transparent reporting form
• Supplementary file 1. Strain table.

### Data availability

The CAD file used to generate the microfluidic device is available on a github repository (https://github.com/TAspert/Continuous_filtration copy archived at https://archive.softwareheritage.org/swh:1:rev:8476f782bf6ff8fff2a9c78172cc8f072cc73916). The source data used to make the panels (excluding raw image files) are included for each figure. Due to size constraints representative raw image data for Figure 1 is available at Zenodo (https://doi.org/10.5281/zenodo.5592983) and the remaining raw image data, including files for Figures 2 and 3, are available on request from the corresponding author.

The following dataset was generated:

| Author(s) | Year | Dataset title | Dataset URL | Database and Identifier |
|---|---|---|---|---|
| Aspert T, Jacquel B, Charvin G | 2021 | Dataset pHluorin cells experiencing entry into quiescence | https://doi.org/10.5281/zenodo.5592983 | Zenodo, 10.5281/zenodo.5592983 |

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
