## [Editor Report]

The cell fate program that is set in motion as yeast cells transition from fermentation to respiration is still not well understood. The development of the microfluidic platform described in this manuscript could make a significant contribution to understanding the succession of metabolic and structural changes occurring during this transition. The application of single cell tracking to monitor the temporal program of these changes represents a major technical advance that will be of general interest to researchers interested in defining the developmental programs that contribute to cellular quiescence and longevity.

---

## [Decision Letter]

[Editors' note: this paper was reviewed by Review Commons.]

**Decision letter after peer review:**

Thank you for submitting your article "Monitoring single-cell dynamics of entry into quiescence during an unperturbed lifecycle" for consideration by *eLife*. Your article has been reviewed by 2 peer reviewers at Review Commons, and the evaluation at *eLife* has been overseen by Bavesh Kana as the Senior and Reviewing Editor.

Based on your manuscript, the reviews and your responses, we invite you to submit a revised version incorporating the revisions. When preparing your revisions, please also address the following points:

The reviewers concur that your work represents an important breakthrough and remain enthusiastic about the study. However, reservations about BY4741 remain. It is true that these strains are very commonly used, but it is also true that their growth defects post-diauxie were recognized and published over a decade ago (pmid 21120607, 22780918). Importantly, these defects are specific to BY4741/4742 and not shared by S288c. Reviewers request that you conduct a growth curve comparison with the BY strains and W303 prototrophs. If these strains display comparable growth beyond the diauxic shift, this observation should resolve the matter.

---

## [Author Response]

The reviewers concur that your work represents an important breakthrough and remain enthusiastic about the study. However, reservations about BY4741 remain. It is true that these strains are very commonly used, but it is also true that their growth defects post-diauxie were recognized and published over a decade ago (pmid 21120607, 22780918). Importantly, these defects are specific to BY4741/4742 and not shared by S288c. Reviewers request that you conduct a growth curve comparison with the BY strains and W303 prototrophs. If these strains display comparable growth beyond the diauxic shift, this observation should resolve the matter.

To address potential issues associated with the use of the BY4741 strain, we have run complementary experiments as follows.

1. We have performed population growth curves with a variety of strains to test their ability to switch to respiration after the diauxic shift. For this, we have used both auxotroph (BY and W303) strains, and prototrophs from several related origins (FY, S288C, and BY). Our results indicate that both auxotrophs and prototrophs are able to undergo a bona fide switch to respiration. Even though the BY auxotroph may experience a more pronounced delay in switching to respiration compared to the W303 strains, the final density of cells is identical (new Figure 1—figure supplement 2).

Importantly, we have checked that all the auxotrophies of the strains used in this assay were correct. In addition, we have used a FY Rho^0^ strain (i.e., a strain unable to respire) as a negative control to assess the ability of the other strains to switch to respiration. Last, we have looked for a W303 prototroph in the literature as well as in our own stock but we failed to find one. We have asked Joseph Schacherer (Université of Strasbourg), a renowned expert in *S.cerevisae* phylogenetics, who confirmed that this lab strain was derived from an auxotrophic ancestor.

Altogether, this new dataset indicates that the single cell analyses performed in our study with a BY prototroph are likely to apply to other backgrounds, whether auxotroph or prototroph. We have included a description of these experiments in the main text, and we have justified these experiments by mentioning the potential concern with the BY strain.

2. We have performed additional single-cell experiments with a W303 strain that carries the internal pH-marker pHluorin2 to compare the results to those obtained in the BY background. Our results indicate that the W303 cells display the same dynamics as observed in the BY background (new Figure 1—figure supplement 3). In addition, similarly as in BY cells, a small fraction of the W303 cells fail to switch to respiration. We believe that these results further confirm the generality of our observations.

Reviewer #1:Evidence, reproducibility and clarity (Required)These investigators have teamed up to solve a technical problem that has thwarted efforts to get a clear picture of the chronology of events in yeast cultures as they naturally exhaust their nutrient supply. This is a challenge because the time course is long and the density of the culture makes single cell analysis problematic. Previous studies involving abrupt starvation have shown that there is a pH drop when nutrients are eliminated, but abrupt starvation also leads to rapid loss of viability compared to what is observed with cells as they respond and adapt to changes in their environment. This microfluidic devise and an intracellular pH detector allowed them to follow pH change as cells transition from fermentation to respiration and stationary phase. About 15% of the population responds completely differently than the other 85%, making this single cell analysis crucial. It also provides a negative control of sorts, to further substantiate the correlations they draw. This 15% fails to enter the respiratory phase and dies rapidly. The pH also drops rapidly and is correlated with loss of mitochondrial function and aggregation of proteins. The 85% of cells that succeed in shifting to respiration suffer the same pH drop, but it is much slower and is correlated with slower protein aggregation, P Body, actin body, and proteosome storage Granule assembly. They also followed the cytoplasmic transition to a glassy state, based on the mobility of protein foci and lipid droplets. This transition occurs at the same pH in both populations but with completely different timing. This recapitulates the transition observed after abrupt starvation. It shows that the same transition occurs in viable, quiescent cells and provide further evidence that it is correlated with pH changes.The only concern I have is that they used only one strain, which reduces the universality of their findings. Moreover, it is ambiguous which strain was used. The strain table says they used S288c which is known to carry a hap1 mutation that compromises respiration and isn't the best choice for studying respiring cells. The text mentions that they are working in the BY background, where most of their GFP studies have been carried out. The BY strains have the same hap1 mutation and several other unknown polymorphisms that preventethanol utilization and biomass increase after the diauxic shift.

To address the point raised by reviewer #1, we used a BY strain (from an S288C background), which is one of the most widely used lab strains in yeast genetics. Although a mutation in the HAP1 gene was identified in this strain, it does not compromise the ability of the cells to respire, as shown in our experiments performed in a non-fermentative medium in this manuscript (Figure 2). Similarly, we have already published growth curves, drop tests, and proliferation rates of this strain in glycerol/ethanol or lactate-containing medium (please see Jimenez et al., JCS, 2014). More importantly, detailed respiration parameters have been measured for our BY background by the Daignan-Fornier lab, a long-standing Sagot lab collaborator (please see Gauthier et al., Mol Mic, 2008). Last, we have previously shown that upon quiescence establishment, there is no difference in mitochondrial network reorganization between BY/S288C and W303 or CEN PK background (Laporte et al., *eLife*, 2018, Figure 2 SupFig1 F-G). Altogether, these results indicate that using this particular lab strain background does not detract from the generality of the observations reported in our manuscript. We will include a short discussion on this specific topic in the main text in a full revision.

Significance (Required)This kind of single cell analysis is clearly the way forward and will have many further applications to understanding how cells adapt to their environment. The paper is well written and the figures are well laid out and easy to understand. It is a significant advance for the field and will set the bar for future experiments. However, this work was done with a single strain that is known to be defective in respiration. It would be extremely valuable to know if their results with this strain are generalizable to other lab and wild strains.Reviewer #2:Evidence, reproducibility and clarity (Required)In this manuscript, the authors interrogated single cells of yeast as they developed into quiescence after the natural depletion of glucose from the culture medium. To do so, they constructed a microfluidic platform to track individual cells in a batch culture of cells transitioning from a growing phase into stationary phase. They then used a number of assays to monitor the metabolic changes that accompany this transition. They observed that internal pH dropped during development of quiescence, with some cells showing a rapid drop and others showing a delayed and heterogeneous drop. At diauxie, cells transition from fermentation to respiration, and the minority of cells that showed a rapid drop in pH were respiration-deficient (R-) and unable to resume growth, while cells with a delayed pH drop were respiration proficient (R+) and able to resume growth. Using established markers to follow the previously described structural changes that accompany the development of quiescence, they found that the pH changes were temporally related to these structural changes. They suggest that the dynamic changes in intracellular pH promote waves of structural remodeling that eventually leads to a transition of cells to a gel-like state. The early drop in pH in R- cells was proposed to lead to a precocious transition of the cytoplasm, contributing to the inability of these cells to resume growth.This study is well done, well-written, and the results are clearly presented and generally convincing. Data accumulation and analysis were well documented and appropriate statistics were used. While the authors provided details of the materials used to construct the microfluidic device, it would be appropriate for them to provide a detailed blueprint (and a video, for example) to other investigators who would like to employ this device in their studies of quiescence once this manuscript has been published.

Regarding reviewer #2’s comments, we are committed to providing all necessary details to fully replicate the microfluidic devices upon a final revision of our manuscript to make it widely accessible to the research community.

Significance (Required)The cell fate program that is set in motion as yeast cells transition from fermentation to respiration is still not well understood. The development of the microfluidic platform described in this manuscript could make a significant contribution to our understanding of the succession of metabolic and structural changes occurring during this transition. The Sagot lab has made a series of important contributions in this area, and the application of single cell tracking to monitor the temporal program of these changes represents a major technical advance that will be of general interest to researchers interested in defining the developmental programs that contribute to cellular quiescence and longevity.